# Preoperative Diaphragm Muscle Atrophy Increases the Likelihood of Postoperative Pulmonary Complications After Lung Cancer Resection: A Pilot Study

**DOI:** 10.3390/cancers17030373

**Published:** 2025-01-23

**Authors:** Janusz Kocjan, Mateusz Rydel, Damian Czyżewski, Mariusz Adamek

**Affiliations:** 1Private Clinic Centre Specializing in Treating of Diaphragm Disorders, Diaphragm Concept Academy, 32-300 Olkusz, Poland; 2Department of Thoracic Surgery, Faculty of Medicine with Dentistry Division, Medical University of Silesia, 40-055 Katowice, Poland; mateusz.rydel@wp.pl (M.R.); damian.czyzewski@o2.pl (D.C.); m.adamek@e.pl (M.A.); 3Department of Radiology, Faculty of Health Sciences with Institute of Maritime and Tropical Medicine, Medical University of Gdansk, 80-210 Gdansk, Poland

**Keywords:** PPCs, lung cancer, diaphragm, atrophy

## Abstract

In this study, whether preoperative diaphragm dysfunction increases the risk of postoperative lung complications (PPCs) after lung cancer surgery was examined. The results showed that patients with diaphragm atrophy had a higher risk of PPCs and longer hospital stays compared to those with normal diaphragm thickness. There was also a moderate link between diaphragm thickness during expiration and the number of complications. These findings suggest that diaphragm atrophy is an important, modifiable risk factor for PPCs, highlighting the potential benefit of early therapeutic interventions to improve outcomes after lung cancer surgery.

## 1. Introduction

Surgical resection of tumor tissue remains the preferred option for curative treatment in the early stages of non-small cell lung cancer [1]. However, the impact of lung surgery on respiratory system function carries a risk of developing postoperative pulmonary complications (PPCs), which increases the likelihood of postoperative mortality and affects long-term patient survival [2,3]. The reported incidence of PPCs after lobectomy ranges from 15% to 30% [4], with an estimated 30-day mortality rate between 1% and 3% [5]. According to a study presented by Baar et al., the presence of two PPCs was associated with a significantly increased mortality rate of 7%, while three or more PPCs increased the mortality rate to 33%, compared to patients with none or only one PPC [6]. Several studies have shown that factors such as race, age, smoking, COPD, and preoperative pulmonary function are associated with a higher risk of PPCs [7,8,9,10]. Some studies have also suggested that the likelihood of PPCs is inversely related to the distance between the surgical incision and the diaphragm, making patients undergoing thoracic surgeries particularly vulnerable [11].

The diaphragm is the primary muscle responsible for breathing, separating the thoracic and abdominal cavities, and generating approximately 60–70% of the tidal volume. Diaphragm dysfunction, in the form of atrophy, leads to decreased contractile strength, reduced endurance, and quicker fatigue. These changes can reduce tidal volume and the ability to sustain effective ventilation, leading to hypoxemia or hypercapnia [12]. The adverse impact of postoperative diaphragmatic dysfunction on PPCs has been reported in several studies following thoracic surgery [13,14,15]. Similar findings have also been observed after cardiac surgeries [16,17,18,19,20]. However, these studies have primarily focused on the impact of postoperative diaphragm dysfunction on the development of PPCs, emphasizing diaphragm excursion while giving limited attention to diaphragm thickness measurements and the diagnosis of diaphragm atrophy. Considering that earlier research has reported a high prevalence of decreased diaphragm muscle mass in patients undergoing lobectomy [21], it is reasonable to explore the influence of preoperative diaphragm dysfunction on postoperative outcomes.

The purposes of the present study were as follows: (I) to investigate the potential relationship between the presence of preoperative diaphragm muscle dysfunction and the risk of PPC occurrence; (II) to assess the impact of diaphragm muscle dysfunction on postoperative outcomes; and (III) to evaluate whether diaphragm muscle dysfunction can affect postoperative blood parameters such as erythrocyte count, hemoglobin level, and hematocrit. We hypothesized that preoperative diaphragm muscle dysfunction could predispose patients to a higher incidence of PPCs and lead to worse postoperative outcomes compared to patients with normal diaphragm muscle function. We also hypothesized that reduced diaphragm thickness may be associated with higher levels of red cells and hemoglobin in the blood.

## 2. Materials and Methods

### 2.1. Study Design and Settings

This was a single-center observational study conducted at the Prof. S. Szyszko Teaching Hospital No 1 of the Silesian Medical University in Zabrze (Poland). Written informed consent was obtained from all participants before the preoperative evaluation. Ethical approval for this study was provided by the local Ethics Committee of Silesian Medical University, Katowice, Poland (Decision number: BNW/NWN/0052/KB1/42/I/24). The examination was conducted in accordance with the Declaration of Helsinki.

### 2.2. Participants

Patients scheduled for anatomical lung lobe resection due to non-small cell lung cancer (TNM stages IA–IIIB) at the Thoracic Surgery Department were screened for enrolment. Patient selection was conducted every Monday due to the limited availability of the diaphragm ultrasonographer. Eligible patients were those aged 18 years or older with pathologically confirmed lung cancer. Exclusion criteria included refusal to participate, self-reported pregnancy, a history of previous thoracic and/or abdominal surgeries, qualification for bilobectomy, segmentectomy, or pneumonectomy, conversion from VATS to open thoracotomy during surgery, physical or mental impairments, poor diaphragm visibility on ultrasound, and inability to perform breathing maneuvers during the ultrasound examination.

### 2.3. Diaphragm Ultrasound Protocol

The thickness of both hemidiaphragms was measured in each patient the day before surgery by a well-trained sonographer with extensive experience in using the ALOKA ultrasound machine. Examinations were conducted at the bedside with patients in a supine position to reduce variability and improve reproducibility. A linear transducer emitting a high-frequency (6–12 MHz) beam was placed at the point where the muscle fibers contact the rib cage, between the 8th and 10th intercostal spaces along the mid-axillary line. B-mode ultrasound, which provides a two-dimensional cross-sectional view of a three-dimensional structure, was selected for imaging. In this view, the diaphragm appeared as a three-layer structure, with one hypoechoic inner muscle layer bordered by two hyperechoic outer membranes (the peritoneum and pleura) [22]. Diaphragm thickness was evaluated during maximal inspiration (ThIns) and maximal expiration (ThExp) phases. Each measurement was taken three times, and the averages were calculated for statistical analysis.

### 2.4. Diaphragm Dysfunction Assessment

Based on the obtained inspiratory and expiratory values, we investigated the potential presence of the following diaphragm dysfunctions:

Diaphragm Atrophy: Standardized diaphragm muscle atrophy is typically defined in the literature as an expiratory thickness of less than 2 mm [23]. In the present study, we adjusted for sex differences by using cut-off values of 1.4 mm for women and 1.9 mm for men, based on the Carillo-Esper study [24]. Given the advanced age of the patients qualified for surgery and decline in muscle mass along with the age, we believe this criterion should most accurately identify patients with diaphragm atrophy.

Diaphragm Paralysis: To assess diaphragm paralysis, we calculated the diaphragm thickness fraction (DTF) index, which reflects the magnitude of diaphragm effort. The DTF was calculated using the following formula: (ThIns − ThExp)/ThExp × 100%. Percentage values below 20% were interpreted as indicative of diaphragm paralysis [25].

Diaphragm Weakness: Diaphragm weakness was expressed as the diaphragm thickening ratio (DTR) index, defined as the difference between end-inspiratory and end-expiratory thickness values [26].

### 2.5. Surgical Procedures

All patients were operated on by one of three dedicated thoracic surgeons with extensive experience working in our department. The surgical approach, either a lateral thoracotomy or a video-assisted thoracoscopy surgery (VATS), was chosen non-randomly by the surgical team performing the lobectomy based on the clinical picture and attributes such as tumor size, patient age, pulmonary function, and the patient’s general condition. Both procedures involved an anatomical resection of the lung lobe where the tumor was located (lobectomy) and the removal of mediastinal lymph nodes (lymphadenectomy).

All patients were placed in the lateral decubitus position under general anesthesia (Propofol, Fentanyl, Sevoflurane) and intubated with a double-lumen tube, allowing for selective ventilation of the lung on the side opposite to the operated lung. VATS resection was performed using a single-port, non-rib-spreading, non-trocar technique. A 3–5 cm single incision was made in the 5th intercostal space, through which the scope camera and all endoscopic tools were introduced into the pleural cavity. The same incision was used to insert pleural drainage at the end of the operation. Lobe resection via thoracotomy was performed through a muscle-sparing posterolateral incision in the 5th or 6th intercostal spaces, using a 15–20 cm lateral skin incision. The serratus anterior muscle was spared, and a rib spreader was used to gain access to the thoracic cavity. After the procedure, two drains were placed in the pleural cavity through separate 2–3 cm incisions.

Once the lobectomy was completed, the thoracic cavity was closed in layers, and an aseptic dressing was applied. Patients were then transported to an intensive care unit for observation for a few hours before being transferred to the thoracic surgical ward. Drains were removed when the lung was fully expanded, air leakage was less than 20–40 mL/h, and daily drainage was less than 300–400 mL.

### 2.6. Data Collection

#### 2.6.1. Preoperative and Intra-Operative Data

Demographic patient characteristics (age, sex) and medical history (TNM stage, type of surgery approach, operated side, resected lobe, previous surgeries) were collected by review of electronic medical hospital records.

#### 2.6.2. Post-Operative Data

The length of hospital stay was calculated based on the number of midnights stayed in the hospital. The drainage time was recorded from the day of the operation until the day when the chest tube was removed. Blood gas analyses (pH, paCO_2_, HCO_3_, Be) were performed on the morning after surgery using a point-of-care blood gas analyzer. Additional laboratory measurements, including erythrocyte count, hemoglobin concentration, and hematocrit level, were performed on a separate blood sample.

#### 2.6.3. Postoperative Pulmonary Complications (PPCs)

Postoperatively, each patient was assessed daily during their hospital stay by their attending physician, who was not involved in the project. A review of clinical records to assess the presence of PPCs was performed by one of the study authors, who did not have access to ultrasound measurement data.

From the wide spectrum of PPCs, in this paper, patients were considered to have PPCs if one or more of the following conditions, based on the European Perioperative Clinical Outcome (EPCO) definition [27], were detected during their hospital stay: pneumonitis, atelectasis, respiratory failure, respiratory infection, bronchospasm, pneumothorax, hypoxemia, or pleural effusion. Additionally, we documented the occurrence of the following postoperative respiratory interventions: noninvasive ventilation, re-intubation with mechanical ventilation, and unplanned postoperative transfer to the ICU.

### 2.7. Statistical Analysis

Data obtained from the present study were presented as number (*n*) and percent (%) or mean (M) and standard deviation (SD) and compared using chi-square test, or Student’s *t*-test. Correlation analysis was performed using the Pearson test. All statistical analysis was performed using the SofaStat Software (version 1.5.7). The *p*-values < 0.05 were considered statistically significant.

## 3. Results

During the study period, 51 patients were screened for eligibility. A total of three patients were excluded during the medical interview due to a history of thoracic (*n* = 2) or abdominal surgery (*n* = 1). Of the remaining patients, 48 completed ultrasonography measurements and were deemed eligible for lung cancer resection via VATS lobectomy or conventional thoracotomy. Following this, two patients were excluded due to conversion from VATS to open thoracotomy during surgery, and one patient did not undergo surgery. This left a final sample of 45 subjects (20 women and 25 men) for analysis. The mean age of the subjects was 64.3 ± 13.72. The detailed clinical characteristics of the patients are presented in Table 1.

Of the 45 cases, 71.1% patients developed at least one postoperative pulmonary complication, while 22.2% patients experienced two or more PPCs. In 28.9% subjects, no PPCs were observed. Detailed data are shown in Figure 1.

The mean number of PPCs was 1.34 ± 1.73, and the type of surgical procedure did not significantly differentiate (*p* = 0.683) between the subgroups undergoing VATS (1.31 ± 0.47) and thoracotomy (1.25 ± 0.44). No cases of pulmonary edema, respiratory failure, or unplanned postoperative transfer to the ICU were observed. Furthermore, no deaths were recorded during the patients’ hospital stay. The most common postoperative complications were pleural effusion (*n* = 23), pneumothorax (*n* = 16), and hypoxemia (*n* = 7). The frequency of other complications was as follows: mechanical ventilation (*n* = 4), pneumonitis (*n* = 4), atelectasis (*n* = 3), bronchospasm (*n* = 3), re-intubation (*n* = 3), respiratory infection (*n* = 2). The incidence of PPCs, divided by procedure type, is shown in Figure 2.

The mean inspiratory and expiratory thicknesses were 2.64 ± 0.07 mm and 1.95 ± 0.04 mm, respectively, with a mean DTF of 53.41 ± 38.86 and a mean DTR of 1.53 ± 0.38. We identified preoperative diaphragm dysfunction in 95.5% (*n* = 43) of 45 patients. The majority of cases involved diaphragm weakness (68.8%, *n* = 31), followed by diaphragm atrophy (42.2%, *n* = 19) and diaphragm paralysis (22.2%, *n* = 10). Atrophy and paralysis were more frequently observed in females (atrophy: 50% [*n* = 10] vs. 36% [*n* = 9], *p* = 0.293; paralysis: 30% [*n* = 6] vs. 16% [*n* = 4], *p* = 0.196), while weakness was more common among males (72% [*n* = 18] vs. 65% [*n* = 13], *p* = 0.953). No significant differences in preoperative characteristics were identified between patients with and without dysfunction, particularly regarding mean age (atrophic: 65.7 ± 15.0 vs. non-atrophic: 64.0 ± 12.5, *p* = 0.917; paralyzed: 63.4 ± 11.2 vs. non-paralyses: 64.8 ± 14.2 *p* = 0.364; weakness: 64.5 ± 11.0 vs. normal strength: 63.0 ± 13.4 *p* = 0.572) and the surgical (VATS vs. thoracotomy) approach distribution (atrophy: 36.8% [*n* = 7] vs. 46.1% [*n* = 12], *p* = 0.606; paralysis: 15.8% [*n* = 3] vs. 26.9% [*n* = 7], *p* = 0.374; weakness: 63.1% [*n* = 12] vs. 73.1% [*n* = 19], *p* = 0.477).

Patients with preoperative diaphragm atrophy were more likely to develop PPCs compared to those with non-atrophic diaphragm. PPCs occurred in 16 of 19 patients with atrophy (84.2%) and 16 of 26 patients without atrophy (61.5%). Regarding specific complications, diaphragmatic atrophy was associated with a higher risk of atelectasis, pleural effusion, and post-surgical re-intubation. No significant differences in the occurrence of PPCs were observed between patients with weak and normal diaphragm strength, or between those with paralyzed and non-paralyzed diaphragms (Table 2).

A single type of diaphragm dysfunction was detected in 68.9% (*n* = 31) of patients, while two types and all three types were observed simultaneously in 20% (*n* = 9) and 11.1% (*n* = 5) of patients, respectively. PPCs developed independently of the number of diagnosed diaphragm muscle dysfunctions in the patients (Figure 3), but the average number of complications was associated with the number of diagnosed diaphragm dysfunctions (one dysfunction: 1.0 ± 1.3, two dysfunctions: 0.8 ± 0.4, three dysfunctions: 4.2 ± 2.8, *p* = 0.013).

The type of surgical procedure did not significantly affect the development of PPCs, as well as not differentiating the mean number of PPCs. Detailed findings are presented on Table 3.

We demonstrated a moderate, statistically significant correlation (r = −0.356, *p* = 0.0164) between diaphragm expiratory thickness and number of PPCs (Figure 4). This relationship was observed in patients who underwent both VATS [r = −0.306, *p* = 0.0492] and thoracotomy [r = −0.419, *p* = 0.0329] (Figure 5). However, no significant association was found with inspiratory thickness (r = −0.299, *p* = 0.0758). Similarly, the values of the DTF index (r = 0.056, *p* = 0.7172) and DTR (r = 0.101, *p* = 0.6924) did not influence the number of PPCs.

The impact of diaphragmatic atrophy on post-surgical variables is presented in Figure 6. Patients with an atrophic diaphragm were characterized by longer hospital stays (*p* = 0.0362) and required drains for more days (6.9 ± 7.0 vs. 4.2 ± 2.3, *p* = 0.0622) compared to those with normal muscle thickness. No significant differences were observed between patients with a paralyzed diaphragm and those without paralysis (drains: 6.7 ± 8.7 vs. 4.8 ± 3.4, *p* = 0.3096; hospital stay: 12.6 ± 10.0 vs. 8.1 ± 3.5, *p* = 0.0289), or between patients with a diaphragm weakness and those with normal diaphragm strength (drains: 5.4 ± 5.7 vs. 4.9 ± 3.0, *p* = 0.764; hospital stay: 9.5 ± 6.6 vs. 8.1 ± 3.4, *p* = 0.4705), with respect to these two analyzed variables. The type of surgical procedure had no impact on postoperative outcomes (Table 4).

Patients with diaphragm atrophy exhibited a higher number of erythrocytes and increased partial pressure of CO₂ within the first 24 h after the procedure. However, these differences were not statistically significant. Detailed data on postoperative blood parameters and gasometry measurements are presented in Table 5.

## 4. Discussion

In this study, the key findings are as follows: (1) of the three types of possible diaphragm muscle dysfunctions, only the presence of preoperative diaphragm atrophy is associated with a higher risk of PPCs occurrence; (2) the value of diaphragm expiratory thickness shows a moderate relationship with the number of PPCs; and (3) preoperative diaphragm ultrasound may be a useful screening tool for detecting patients at an increased risk of developing PPCs. Additionally, preoperative respiratory physiotherapy aimed at restore the diaphragm mass should be implemented for these patients prior to thoracic surgery.

To date, many studies have explored the etiology, risk factors, and prevention strategies associated with postoperative pulmonary complications following various surgical procedures. Some studies have identified post-surgical diaphragm dysfunction as a contributing factor to the development of PPCs [13,14,15,16,17,18,19,20]. However, preoperative diaphragm dysfunction has been recognized as a critical determinant of postoperative pulmonary outcomes in only a few studies. In one of them, the authors found that diaphragmatic dysfunction increases the susceptibility to PPCs following radical resection of esophageal cancer in elderly patients [28]. Another prospective cohort study conducted in adults demonstrated that a low preoperative diaphragm thickness fraction identifies the high-risk PPC patients undergoing curative digestive cancer surgery [29]. Patients with reduced preoperative diaphragm function also experienced higher rates of complications, such as pneumonia, prolonged mechanical ventilation, and the need for re-intubation after cardiac surgery [30]. A similar observation that a low preoperative diaphragm thickness fraction was associated with an increased incidence of PPCs was noted after robot-assisted laparoscopic prostatectomy [31]. Additionally, preoperatively increased diaphragm echodensity is related to the occurrence of PPCs in patients undergoing major abdominal surgery [32]. To the best of our knowledge, this is the first study designed to directly assess the impact of preoperative diaphragm muscle atrophy on the risk of PPCs occurrence in patients underwent lung cancer resection.

The data presented in Table 2 indicate a significantly higher incidence of postoperative complications in patients with preoperative diaphragmatic atrophy compared to those with normal diaphragmatic thickness. This association was evident both in the higher likelihood of patients with diaphragmatic atrophy to develop multiple types of PPCs, as well as in specific types of PPCs, such as pleural effusion, atelectasis, and the need for postoperative re-intubation. In our opinion, diaphragmatic atrophy does not directly cause PPCs but predisposes patients to their development by disrupting optimal breathing mechanics. By reducing diaphragmatic mobility and postoperative respiratory effort, it compromises lung expansion and induces an inefficient respiratory pattern. This leads to incomplete ventilation in certain lung regions, increasing the risk of atelectasis. Furthermore, the diaphragm plays a crucial role in lymphatic drainage within the thoracic cavity. Diaphragmatic atrophy can impair this drainage process by altering pressure gradients, thereby hindering the clearance of fluids and secretions from the lungs. This dysfunction may also contribute to fluid accumulation in the pleural space, further exacerbating respiratory complications.

In a previous study, patients with diaphragmatic dysfunction, identified by an excursion value of less than 1 cm at 24 h postoperatively, experienced a higher incidence of postoperative pulmonary complications compared to those without this dysfunction [33]. In our study, we found no correlation between preoperative DTF index values (which indirectly reflects the magnitude of diaphragm contraction) and the frequency of complications. Based on our findings and the studies mentioned above, it appears that only the presence of post-surgery diaphragmatic paralysis serves as a significant risk factor for PPCs. This thesis is supported by other previous studies that reported significant increases in diaphragmatic paralysis after VATS lobectomy and thoracotomy compared to preoperative levels [21].

In the present paper, we assumed that reduced ventilatory efficiency and impaired oxygen exchange in the lungs, resulting from diaphragm dysfunction, could lead to an increase in red blood cell count and hemoglobin as a compensatory mechanism to enhance oxygen delivery. However, the data presented in Table 4 did not support this hypothesis. Although we observed slightly higher values of erythrocyte count and PaCO_2_ levels in patients with diaphragm atrophy compared to those with normal diaphragm thickness, the differences were not statistically significant. However, it should be noted that in our study, the blood gas measurements were performed only on the first day after surgery and were not repeated later during the hospital stay. Therefore, it cannot be ruled out that hypercapnia and increased erythropoiesis may manifest at a later stage. Addressing this issue requires further studies with a longer duration of patient observation.

Our study was conducted on individuals with relatively poor diaphragmatic function before to surgery. Almost all patients were diagnosed with at least one of three types of diaphragm dysfunction. This observation could explain the high rate of complications reported in this paper. However, it raises an important question about the underlying causes of this condition. We suppose that alterations in breathing patterns, specifically shallow breathing, may lead to reduced muscle mobility and subsequent decreases in muscle mass. Given that most patients with lung cancer are smokers, smoking could potentially contribute to the high prevalence of dysfunctional diaphragms observed in this population. Previous studies have suggested that smoking damages diaphragm muscle fibers and induces diaphragm atrophy. However, whether such damage is reversible remains uncertain [34].

In the present study, most patients underwent open thoracotomy rather than VATS. Recent studies have shown that minimally invasive lobectomy offers significant advantages over standard thoracotomy for early-stage non-small cell lung cancer [21,35,36,37]. However, in our study, the type of surgical procedure did not significantly influence the development of PPCs. First, no differences were observed between the two patient subgroups in terms of the percentage of diaphragm muscle atrophy prior to surgery. Second, in both subgroups, diaphragm atrophy was associated with a higher number of PPCs. Finally, although the percentage of PPCs was slightly higher after thoracotomy compared to VATS, the risk of PPCs remained elevated in patients with diaphragm atrophy compared to those without atrophy in both subgroups. Based on these findings, it appears that diaphragm atrophy is a risk factor for PPC development, independent of the surgical approach.

The strength of our study lies in the comprehensive assessment of preoperative diaphragm muscle function, including the evaluation of all potential types of dysfunction in this area. However, our study has several limitations, primarily its small sample size and single-center design. These factors may have reduced the statistical power of our findings and could limit the generalizability of the results. Consequently, the results and conclusions presented here should be interpreted with caution at this stage. Furthermore, our study did not specifically evaluate the relationship between diaphragm muscle function and PPCs in patients undergoing other types of surgeries (bilobectomy, segmentectomy, wedge resection), which should be investigated in further targeted studies. Despite the sonographer’s many years of experience, having measurements taken by a single individual may limit reproducibility across different settings. Additionally, patient selection was restricted to one day per week due to the limited availability of the ultrasonographer, which may have introduced selection bias by excluding patients who could not be assessed on that specific day. This may have also excluded patients with urgent or unscheduled surgeries. And finally, we excluded patients who underwent conversion from VATS to thoracotomy to maintain homogeneity within each surgical group, allowing for more accurate comparisons and minimizing confounding variables. However, it is important to consider that these excluded patients may have had unique clinical characteristics or complications not accounted for in this study.

## 5. Conclusions

In our study, we demonstrated that patients with diaphragm atrophy are at a higher risk of developing postoperative pulmonary complications and experience longer postoperative hospital stays, independent of the type of surgery, compared to those with normal preoperative diaphragm thickness. The findings presented here may provide valuable insights for predicting and screening high-risk patients for PPCs after lobectomy for lung cancer. Crucially, unlike other predictive factors such as age, COPD, or smoking, diaphragmatic atrophy represents a modifiable risk factor that can potentially be addressed through early therapeutic intervention. In this context, prehabilitation, particularly in the form of diaphragm-strengthening exercises, could improve patient outcomes and should be integrated into a multidisciplinary approach for resectable lung cancer patients. Further large-scale, randomized studies comparing usual care with individualized exercise-based strategies are necessary to optimize care for thoracic surgery patients.

## Figures and Tables

**Figure 1 cancers-17-00373-f001:**
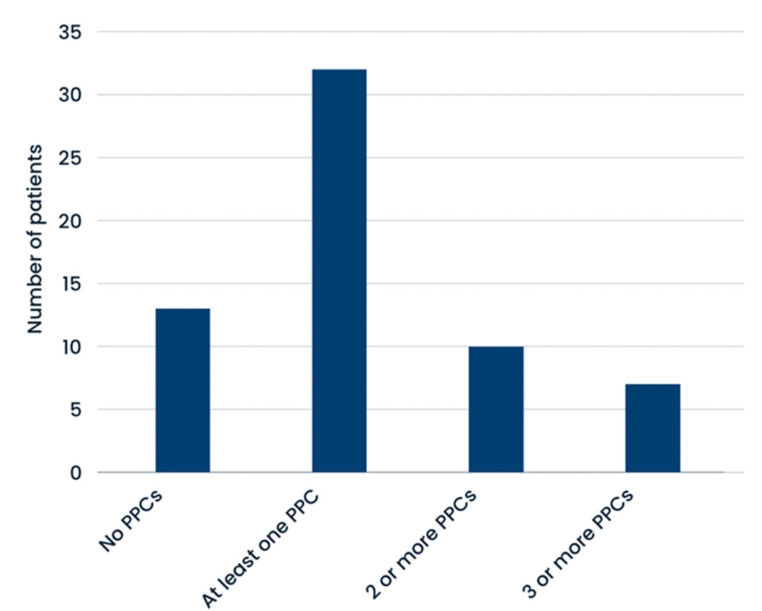
Number of patients with PPCs.

**Figure 2 cancers-17-00373-f002:**
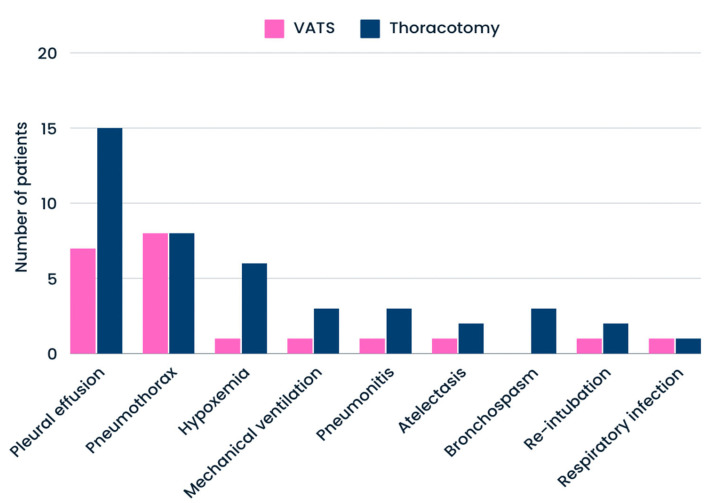
Specific incidence of PPCs.

**Figure 3 cancers-17-00373-f003:**
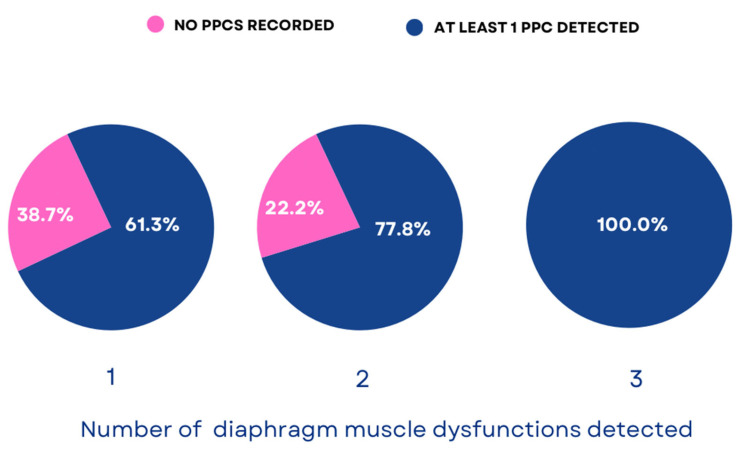
The percentage of postoperative complications based on the number of diagnosed diaphragm dysfunctions.

**Figure 4 cancers-17-00373-f004:**
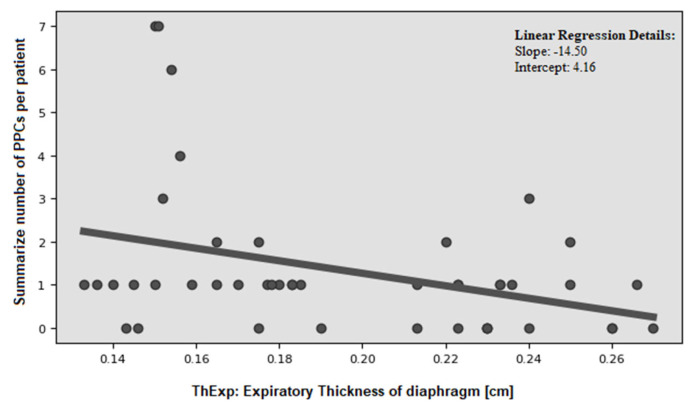
Correlation between diaphragm expiratory thickness and the occurrence of PPCs.

**Figure 5 cancers-17-00373-f005:**
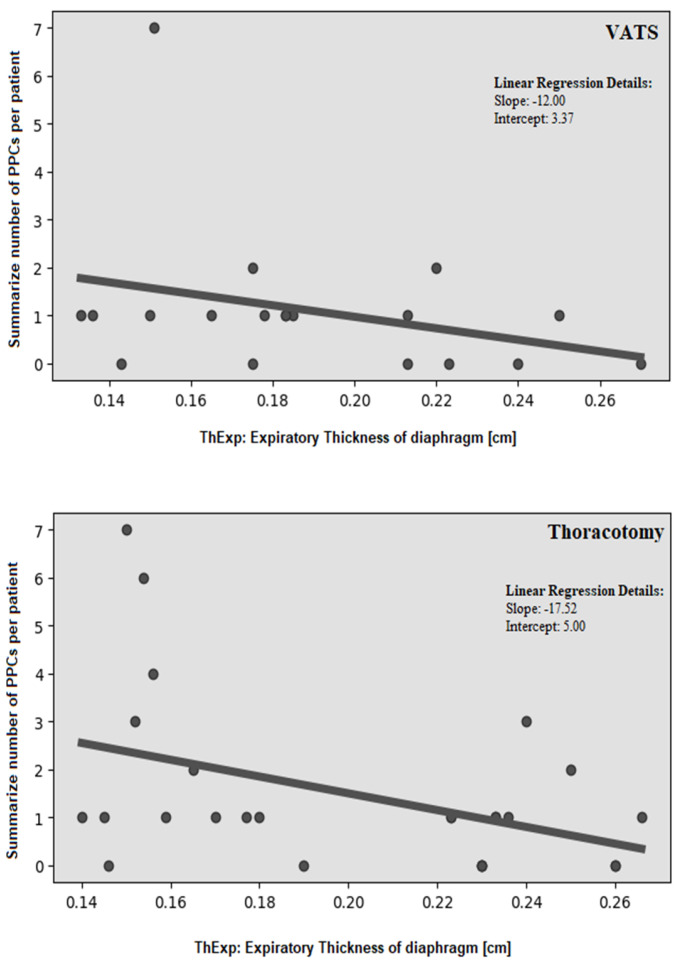
Correlation between diaphragm expiratory thickness and the occurrence of PPCs among patients undergone VATS and Thoracotomy.

**Figure 6 cancers-17-00373-f006:**
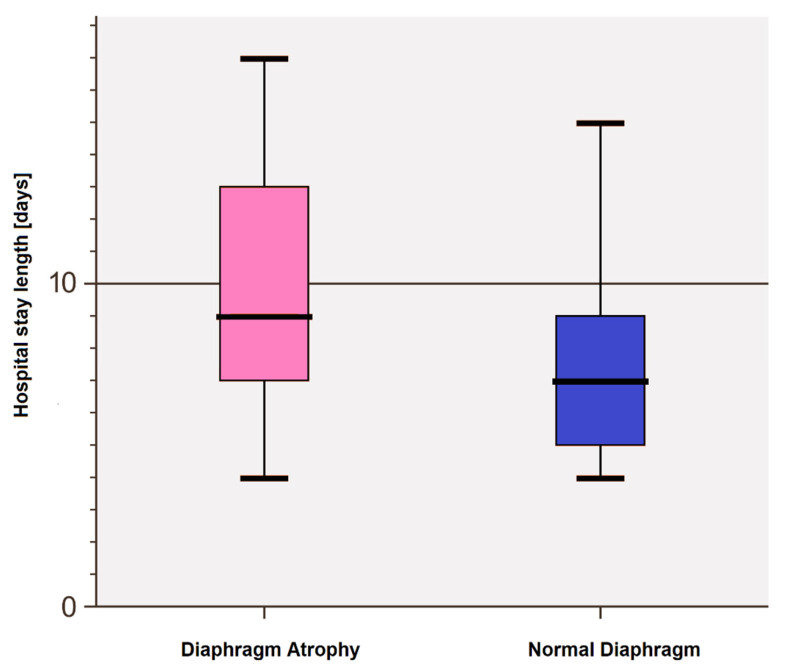
Differences in the length of hospital stay after surgery due to the presence or absence of diaphragm muscle atrophy.

**Table 1 cancers-17-00373-t001:** Clinical characteristics of participants.

Variables	N (%) or Mean ± SD
Surgical approach:	
VATS	19 (42.2%)
Conventional thoracotomy	26 (57.8%)
Involved side:	
Left	21 (53.3%)
Right	24 (46.7%)
Resected lobe:	
Right upper	11 (24.4%)
Right middle	3 (6.7%)
Right lower	11 (24.4%)
Left upper	8 (17.8%)
Left lower	12 (26.7%)
Complete radical lymphadenectomy:	
Yes	45 (100%)
No	0 (0%)
TNM staging:	
IA	14 (31.1%)
IB	8 (17.8%)
IIA	4 (8.9%)
IIB	11 (24.4%)
IIIA	8 (17.8%)
Drains [days]:	5.32 ± 4.96
Length of hospital stay [days]:	9.19 ± 5.80
Postoperative blood parameters:	
Erythrocyte [10^6^/uL]	4.50 ± 0.40
Hemoglobin [g/dL]	13.91 ± 1.34
Hematocrit [%]	40.95 ± 3.42
Postoperative blood gas parameters:	
pH	7.4 ± 0.04
PaCo_2_ [mmHg]	37.74 ± 6.09
HCO_3_ [mEq/L]	22.71 ± 2.41
BE [mmol/L]	−1.63 to 2.0

Abbreviations: PaCo_2_ = partial pressure of carbon dioxide; BE = base excess; mmHg = millimeters of mercury; mmol = millimol; L = liter; mEq = milliequivalent.

**Table 2 cancers-17-00373-t002:** Incidence of postoperative pulmonary complications (PPCs) depending on the type of diaphragm muscle dysfunction.

Variables	Diaphragm Atrophy			Diaphragm Paralysis			Diaphragm Weakness		
	Yes (*n* = 19)	No (*n* = 26)	*p*	Yes (*n* = 10)	No (*n* = 35)	*p*	Yes (*n* = 31)	No (*n* = 14)	*p*
**PPCs**									
mean number of PPCs:	2.1 ± 2.2	0.9 ± 1.0	**0.0163**	2.5 ± 2.6	1.0 ± 1.2	**0.0117**	1.0 ± 1.1	1.0 ± 1.1	0.5766
No PPCs (*n* = 13)	3 (15.8%)	10 (37.0%)	**0.0486**	1 (10%)	12 (34.3%)	0.1351	10 (32.3%)	3 (21.3%)	0.4581
at least 1 PPC (*n* = 32)	16 (84.2%)	16 (61.5%)	**0.0486**	9 (90%)	23 (65.7%)	0.1351	21 (67.7%)	11 (78.1%)	0.4581
2 or more PPCs (*n* = 10)	6 (31.6%)	4 (15.4%)	0.1747	4 (40%)	6 (17.4%)	0.0730	7 (22.6%)	3 (21.3%)	0.5196
3 or more PPCs (*n* = 7)	5 (26.3%)	2 (7.7%)	0.0787	3 (30%)	4 (11.4%)	0.0787	5 (16.1%)	2 (14.2%)	0.8995
**Individuals PPCs**									
Atelectasis (*n* = 3)	3 (15.8%)	0 (0.0%)	**0.0327**	0 (0.0%)	3 (7.9%)	0.4416	3 (9.7%)	0 (0.0%)	0.2280
pneumothorax (*n* = 16)	7 (36.8%	9 (33.3%)	0.8057	3 (42.9%)	13 (34.2%)	0.6605	10 (32.3%)	6 (42.6%)	0.4971
respiratory infection (*n* = 2)	2 (10.5%)	0 (0.0%)	0.0847	0 (0.0%)	2 (5.2%)	0.5346	2 (6.4%)	0 (0.0%)	0.3300
bronchospasm (*n* = 3)	2 (10.5%)	1 (3.7%)	0.3561	1 (14.3%)	2 (5.2%)	0.3792	2 (6.4%)	1 (7.1%)	0.9311
pleural effusion (*n* = 23)	13 (68.4%)	10 (37.0%)	**0.0457**	3 (30.0)	20 (57.1%)	0.2963	14 (45.2%)	9 (64.2%)	0.3433
mechanical ventilation (*n* = 4)	3 (15.8%)	1 (3.7%)	0.1520	1 (14.3%)	3 (7.9%)	0.4416	3 (9.7%)	1 (7.1%)	0.7461
pneumonitis (*n* = 4)	3 (15.8%)	1 (3.7%)	0.1520	1 (14.3%)	3 (7.9%)	0.4416	3 (9.7%)	1 (7.1%)	0.7461
re-intubation (*n* = 3)	3 (15.8%)	0 (0.0%)	**0.0327**	0 (0.0%)	3 (7.9%)	0.4416	2 (6.4%)	1 (7.1%)	0.9311
Hypoxemia (*n* = 7)	5 (26.3%)	2 (7.7%)	0.0787	3 (30.0)	4 (11.4%)	0.0787	4 (12.8%)	3 (21.3%)	0.7995

Abbreviations: PPCs—postoperative pulmonary complications.

**Table 3 cancers-17-00373-t003:** The number of PPCs in patients with and without diaphragm atrophy depending on the type of surgical procedure.

Variables	VATS			Thoracotomy		
	DA (*n* = 7)	NDT (*n* = 12)	*p*	DA (*n* = 12)	NDT (*n* = 19)	*p*
PPCs:			0.8295			0.0620
yes	5 (71.4%)	8 (66.7%)		11 (91.7%)	9 (47.4%)	
no	2 (28.6%	4 (33.3%)		1 (8.3%)	6 (31.6%)	
2 or more PPCs:			0.6834			0.2205
yes	1 (14.3%)	1 (8.3%)		5 (41.7%)	3 (15.8%)	
no	6 (85.7%)	11 (91.7%)		7 (58.3%)	12 (84.2%)	
3 or more PPCs:			0.1786			0.2142
yes	1 (14.3%)	0 (0.0%)		4 (33.3%)	2 (10.5%)	
no	6 (85.7%)	12 (100%)		8 (66.7%)	13 (68.4%)	
Mean number of PPCs:	1.71 ± 2.43	0.75 ± 0.62	0.0487	2.33 ± 2.22	1.00 ± 1.19	0.0435

Abbreviations: DA—diaphragm atrophy; NDT—normal diaphragm thickness; PPCs—postoperative pulmonary complications.

**Table 4 cancers-17-00373-t004:** Postoperative outcomes *n* patients with and without diaphragm atrophy depending on the type of surgical approach.

Variables	VATS			Thoracotomy		
	DA (*n* = 7)	NDT (*n* = 12)	*p*	DA (*n* = 12)	NDT (*n* = 19)	*p*
Length of hospital stay [days]	10.6 ± 9.1	7.1 ± 3.3	0.2397	11.7 ± 7.4	8.2 ± 3.2	0.1056
Drains [days]	5.3 ± 3.4	4.0 ± 2.1	0.3221	7.9 ± 8.4	4.3 ± 2.5	0.1298

Abbreviations: DA—diaphragm atrophy; NDT—normal diaphragm thickness. Data are presented as mean ± standard deviation.

**Table 5 cancers-17-00373-t005:** Postoperative blood parameters depending on diaphragm muscle dysfunction type.

Variables	Diaphragm Atrophy			Diaphragm Paralysis			Diaphragm Weakness		
	Yes (*n* = 19)	No (*n* = 26)	*p*	Yes (*n* = 10)	No (*n* = 35)	*p*	Yes (*n* = 31)	No (*n* = 14)	*p*
Erythrocyte	4.63 ± 0.41	4.40 ± 0.34	0.056	4.65 ± 0.24	4.50 + 0.43	0.313	4.58 ± 0.39	4.45 ± 0.43	0.321
Hemoglobin	13.79 ± 1.28	13.98 ± 1.41	0.641	14.04 ± 0.87	13.88 + 1.47	0.758	13.89 ± 1.40	13.99 ± 1.30	0.823
Hematocrit	40.75 ± 3.06	41.09 ± 3.71	0.754	41.57 ± 2.19	40.81 + 3.75	0.548	40.84 ± 3.54	41.27 ± 3.39	0.704
Blood pH	7.39 ± 0.05	7.40 ± 0.03	0.832	7.37 ± 0.04	7.41 + 0.04	0.278	7.39 ± 0.04	7.40 ± 0.04	0.581
PaCO_2_	36.96 ± 5.56	38.76 ± 6.76	0.378	40.70 ± 6.00	36.64 + 5.85	0.071	37.52 ± 6.38	38.31 ± 5.49	0.735
HCO_3_	23.08 ± 1.99	22.43 ± 2.70	0.430	23.03 ± 2.32	22.6 + 2.48	0.637	22.47 ± 2.54	23.36 ± 2.01	0.330
BE	−1.4 ± 1.8	−1.8 ± 2.17	0.582	−2.02 ± 2.19	−1.48 + 1.94	0.477	−1.02 ± 1.67	−1.85 ± 2.09	0.264

Abbreviations: PaCO_2_ = partial pressure of carbon dioxide; BE = base excess. HCO_3_ = bicarbonate.

## Data Availability

The data presented in this study are available upon request from the corresponding author.

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
