# Peer review of "Preoperative Diaphragm Muscle Atrophy Increases the Likelihood of Postoperative Pulmonary Complications After Lung Cancer Resection: A Pilot Study"

_cancers, 2025, doi:10.3390/cancers17030373_

Round 1

Reviewer 1 Report

Comments and Suggestions for Authors

Postoperative pulmonary complications are common risk among patients of lung cancer who have undergone surgical resection for treatment. There are several preoperative factors that could alter these complications. In the present submission by Kocjan et al., the authors have examined preoperative diaphragm muscle dysfunctions and found evidence that atrophy of the diaphragm bears a greater risk in this regard. They recommend patient treatments that could fix the diaphragm thickness before performing the surgery and overall, the results have potential to improve lung cancer surgery outcomes. Their methodology and data analysis are sound, and the author have properly identified the caveats and limitations of their study. I have some comments as outlined below.

1.        Line 189 to 191. The figure/table reference for these results is missing so it is a bit difficult to locate the data.

2.        For Figure 2, a pie chart or a stacked bar graph may provide a better visual of the results.

3.        Line 239 to 241. The figure/table reference for these results is missing.

4.        Discussion paragraph 2 (line 280 to 293): The references in this paragraph all start with the name of the authors. This is not necessary, I would suggest the authors to simply cite the articles and tie the results from those together to convey their conclusion.  

Author Response

Dear Reviewer,

First of all, we would like to thank for thorough and thoughtful review of our manuscript. We greatly appreciate the time and effort you have dedicated to providing detailed feedback and valuable insights. Your comments help us to improve the quality and clarity of the paper. We have carefully considered all your suggestions and have made the necessary revisions to address each point. Below, we presented our answers on your comments

Comments 1: [Line 189 to 191. The figure/table reference for these results is missing so it is a bit difficult to locate the data]

Response 1: [We have presented the data in lines 189-191 in text form only. However, at your suggestion, we have also included them as a chart - Figure 1 in revised manuscript]. 

Comments 2: [For Figure 2, a pie chart or a stacked bar graph may provide a better visual of the results].

Response 2: [Thank you for your suggestion. We have changed the graph to improve data readability.]

Comments 3: [Line 239 to 241. The figure/table reference for these results is missing.]

Response 3: [We have presented this data in text form only to avoid duplicating it in two places, especially that findings were not statistically significant. If the Reviewer would like us to add graphs for this data, please let us know].

Comments 4: [Discussion paragraph 2 (line 280 to 293): The references in this paragraph all start with the name of the authors. This is not necessary, I would suggest the authors to simply cite the articles and tie the results from those together to convey their conclusion.]

Response 4: [Thank you for your comment. We have removed the author name from the discussion]. 

Reviewer 2 Report

Comments and Suggestions for Authors

Thank you for the opportunity to review this important manuscript. Here are my comments and suggestions.

Line 64: Full stop is missing.

Why was ''conversion from VATS to 90 open thoracotomy during surgery'' an exclusion criterion when both open and VATS procedures were included?

The open and VATS procedures should be analyzed separately. One would not expect the same rate of PPC and the speed of respiratory function recovery in patients with 3-5 cm incisions and one drain compared to 15-20 cm incisions and 2 drains.

Author Response

Dear Reviewer,

First of all, we would like to thank for thorough and thoughtful review of our manuscript. We greatly appreciate the time and effort you have dedicated to providing detailed feedback and valuable insights. Your comments help us to improve the quality and clarity of the paper. We have carefully considered all your suggestions and have made the necessary revisions to address each point. Below, we presented our answers on your comments.

Comment 1: [Line 64: Full stop is missing.]

Response 1: [Thank you for noticing this shortcoming. We have corrected it and added a period at the end of the sentence]. 

Comment 2: [Why was ''conversion from VATS to 90 open thoracotomy during surgery'' an exclusion criterion when both open and VATS procedures were included?]

Response 2: [The exclusion of “conversion from VATS to open thoracotomy during surgery” was applied to preserve consistency in surgical approaches. We aimed to ensure that each surgical group remains homogeneous, facilitating more accurate comparisons and avoiding the introduction of confounding variables. Additionally, conversions can reflect complications or technical difficulties that might skew the results. For this reason, converted procedures are more of a hybrid approach and do not fit cleanly into either the VATS or open thoracotomy groups. This could lead to inconsistent data and make it difficult to attribute outcomes to a specific technique].

Comment 3: [The open and VATS procedures should be analyzed separately. One would not expect the same rate of PPC and the speed of respiratory function recovery in patients with 3-5 cm incisions and one drain compared to 15-20 cm incisions and 2 drains.]

Response 3: [Thank you for your comment. We completely agree with your suggestion. In response, we have improved the graph titled „Specific Incidence of PPCs”, which shows the number of recorded PPCs. The data are now presented separately in the figure 2 for patients undergoing VATS and thoracotomy. Data for the overall population are included in the text above the figure. Regarding other analysis variables (length of hospital stay, drainage time), the data were already presented separately for both surgical procedures (Table 3, Table 4, and Figure 4 - now Figure 5 in revised manuscript).]

Reviewer 3 Report

Comments and Suggestions for Authors

The study addresses a critical and modifiable risk factor—preoperative diaphragm atrophy—for postoperative pulmonary complications (PPCs) in lung cancer resection. It is an interesting research. However, the research design and the visualization of findings should be improved.

The study's visualization can be improved by incorporating clear and informative figures with detailed labels, annotations, and accessible color coding to differentiate groups effectively. Replacing or complementing tables with bar graphs, stacked charts, or box-and-whisker plots would make data comparisons more intuitive, particularly for subgroup analyses such as PPC incidence by diaphragm dysfunction type. Scatter plots showing correlations, such as between diaphragm thickness and PPCs, should include regression lines, confidence intervals, and statistical indicators to highlight key trends. Additionally, heatmaps could visualize complex relationships between multiple factors, enhancing clarity. A summary infographic or flowchart could provide an at-a-glance overview of the study's findings, such as the relationship between diaphragm dysfunction and PPCs or suggested prehabilitation pathways. Consistency in visual style, including fonts, line thicknesses, and colors, would further enhance the readability and overall impact of the visualizations.

The study was conducted at a single hospital, which limits the generalizability of the findings. Patient demographics, clinical practices, and healthcare resources may vary significantly across institutions, reducing the broader applicability of the results.

Of the 51 patients initially screened, 6 were excluded (11.8%). While some exclusions (e.g., due to prior surgeries or intraoperative changes) are justified, the relatively high exclusion rate raises concerns about the representativeness of the final cohort. These exclusions could result in selection bias, as the final group may not fully represent all patients undergoing lung cancer resection.

The final sample size of 45 participants is relatively small for deriving statistically robust conclusions, particularly when further subgroup analyses are performed (e.g., diaphragm atrophy vs. no atrophy). This limitation reduces the power to detect significant differences, especially for rarer outcomes.

Patients excluded due to conversion from VATS to thoracotomy might introduce a bias. Such patients could have unique clinical characteristics or complications not accounted for in the study, potentially skewing the observed outcomes in favor of one surgical approach.

Patient selection was restricted to Mondays due to the limited availability of the diaphragm ultrasonographer. This may have introduced a selection bias, potentially excluding patients who could not be assessed on that specific day.

The relatively homogeneous selection criteria (e.g., limited to TNM stages IA–IIIB) exclude other relevant patient groups, such as those with more advanced disease or requiring different surgical procedures. This restricts the study's generalizability to broader patient populations.

Exclusions related to the ability to perform ultrasound measurements (e.g., poor diaphragm visibility) might exclude patients with certain physical or clinical conditions that could independently influence postoperative outcomes, introducing bias.

Patients were selected only on specific days (e.g., Mondays) due to resource constraints (ultrasonographer availability). This practice might inadvertently exclude patients with urgent or unscheduled surgeries, potentially impacting the findings.

Although the sonographer was experienced, the reliance on a single operator and the use of a specific ultrasound machine (ALOKA) might introduce operator and device-related variability, limiting reproducibility across different settings.

The choice of surgical procedure (VATS or thoracotomy) was determined non-randomly by the surgical team. This lack of randomization could introduce confounding factors related to clinical decision-making, such as tumor size or patient condition.

Patients undergoing bilobectomy, segmentectomy, or pneumonectomy were excluded, limiting the study's applicability to a broader range of thoracic surgical procedures.

Postoperative data collection was confined to the immediate hospital stay, which misses potential long-term complications or recovery patterns associated with diaphragm dysfunction.

The final sample size of 45 patients is relatively small, reducing statistical power and increasing the risk of type II errors. This limitation is particularly significant given the study's focus on specific subgroups with different types of diaphragm dysfunction.

Blood gas analysis was conducted only on the morning after surgery. This snapshot approach may not fully capture respiratory changes or complications that develop later during the postoperative period.

Author Response

Comment 1: [The study addresses a critical and modifiable risk factor—preoperative diaphragm atrophy—for postoperative pulmonary complications (PPCs) in lung cancer resection. It is an interesting research. However, the research design and the visualization of findings should be improved.]

Response 1: [We would like to thank you for your thorough and thoughtful review of our manuscript, as well as for identifying the parts that require corrections or modifications. We greatly appreciate the time and effort you have dedicated to providing detailed feedback and valuable insights. Your comments not only help us improve the quality of this paper but will also contribute significantly to improve the final project. Below, we presented our answers on your comments.]

Comment 2: [The study's visualization can be improved by incorporating clear and informative figures with detailed labels, annotations, and accessible color coding to differentiate groups effectively. Replacing or complementing tables with bar graphs, stacked charts, or box-and-whisker plots would make data comparisons more intuitive, particularly for subgroup analyses such as PPC incidence by diaphragm dysfunction type. Scatter plots showing correlations, such as between diaphragm thickness and PPCs, should include regression lines, confidence intervals, and statistical indicators to highlight key trends. Additionally, heatmaps could visualize complex relationships between multiple factors, enhancing clarity. A summary infographic or flowchart could provide an at-a-glance overview of the study's findings, such as the relationship between diaphragm dysfunction and PPCs or suggested prehabilitation pathways. Consistency in visual style, including fonts, line thicknesses, and colors, would further enhance the readability and overall impact of the visualizations.]

Response 2: [Thank you for your comment. We agree that the colors of the graphs may have made it difficult to interpret the results easily. We have corrected this to enhance readability. Following your suggestion, we have also changed the type of graph used for Figure 2. This change not only improves visual clarity but also facilitates the analysis of the data. Additionally, we have supplemented the correlation graphs with logistic regression data. A new graphical abstract was used to summarize study findings.]

Comment 3: [The study was conducted at a single hospital, which limits the generalizability of the findings. Patient demographics, clinical practices, and healthcare resources may vary significantly across institutions, reducing the broader applicability of the results.]

Response 3: [We completely agree with your comment. However, we would like to point out that we identified this as a limitation of our study in the discussion section, specifically in the paragraph addressing study limitations. Our goal is for the final project to be a multi-center study].

Comment 4: [Of the 51 patients initially screened, 6 were excluded (11.8%). While some exclusions (e.g., due to prior surgeries or intraoperative changes) are justified, the relatively high exclusion rate raises concerns about the representativeness of the final cohort. These exclusions could result in selection bias, as the final group may not fully represent all patients undergoing lung cancer resection.]

Response 4: [Thank you for this observation. We acknowledge that the exclusion rate of patients from the study is high. However, in our view, all these exclusions are justified to create the most homogeneous group of patients possible].

Comment 5: [The final sample size of 45 participants is relatively small for deriving statistically robust conclusions, particularly when further subgroup analyses are performed (e.g., diaphragm atrophy vs. no atrophy). This limitation reduces the power to detect significant differences, especially for rarer outcomes.]

Response 5: [Of course, we also agree with this suggestion. In the discussion section, within the paragraph on study limitations, we mentioned that the small sample size reduces the statistical power of our study.]

Comment 6: [Patients excluded due to conversion from VATS to thoracotomy might introduce a bias. Such patients could have unique clinical characteristics or complications not accounted for in the study, potentially skewing the observed outcomes in favor of one surgical approach.]

Response 6: [Thank you for this suggestion. We will definitely incorporate it into the target study. In the presented pilot study, we excluded patients who underwent conversion to ensure that each surgical group remained homogeneous, allowing for more accurate comparisons and avoiding the introduction of confounding variables. Conversions often reflect complications or technical difficulties that might skew the results. For this reason, in our opinion, converted procedures represent a hybrid approach and do not fit neatly into either the VATS or open thoracotomy groups. However, we agree with your observation that such patients may have unique clinical characteristics or complications. We have added this information to the study's limitations. In future studies, the sample should be divided into three subgroups: (I) planned VATS, (II) planned thoracotomy, and (III) conversion from VATS to thoracotomy. In our pilot study, only two patients underwent conversion. Due to the small number of cases, an analysis of this factor is not feasible in this paper.

Comment 7: [Patient selection was restricted to Mondays due to the limited availability of the diaphragm ultrasonographer. This may have introduced a selection bias, potentially excluding patients who could not be assessed on that specific day.]

Response 7: [We agree that this may have introduced some degree of selection bias. Unfortunately, due to the availability of the ultrasonographer, we could only perform measurements on one day per week. We have added this information to the study limitations.]

Comment 8: [The relatively homogeneous selection criteria (e.g., limited to TNM stages IA–IIIB) exclude other relevant patient groups, such as those with more advanced disease or requiring different surgical procedures. This restricts the study's generalizability to broader patient populations.]

Response 8: [Thank you for your comment. 
To ensure valuable results, a homogeneous group of patients undergoing surgery for non-small cell lung cancer was selected. Consequently, the stage of cancer progression was restricted to IA-IIIB, as only patients within this range qualify for surgical treatment. Patients with more advanced lung cancer may be eligible for salvage procedures; however, the limited number of such cases and their significantly below-average treatment outcomes necessitate treating these procedures separately and excluding them from comparisons with standardized surgical procedures. While the preoperative assessment of diaphragmatic atrophy and its impact on postoperative complications in other patient groups remains of interest, the small number of salvage procedures makes it challenging to perform robust statistical analyses. As a result, such an analysis would be limited to a case series.]

Comment 9: [Exclusions related to the ability to perform ultrasound measurements (e.g., poor diaphragm visibility) might exclude patients with certain physical or clinical conditions that could independently influence postoperative outcomes, introducing bias.]

Response 9: [Thank you for this comment. In this study, there were no exclusions due to the inability to visualize the diaphragm during ultrasound examination. However, based on our experience, we know that such situations, although rare, can occur despite the significant expertise of the ultrasonographer. For this reason, we must acknowledge that this limitation may arise, and in cases where the measurement cannot be performed, we must exclude the patient from the study, as it is not possible to replace the measurement with another imaging modality.]

Comment 10: [Patients were selected only on specific days (e.g., Mondays) due to resource constraints (ultrasonographer availability). This practice might inadvertently exclude patients with urgent or unscheduled surgeries, potentially impacting the findings.]

Response 10: [Thank you for your comment. In our clinic, all procedures are planned. The lack of daily availability of the ultrasonographer naturally limits the ability to study urgent cases.]

Comment 11: [Although the sonographer was experienced, the reliance on a single operator and the use of a specific ultrasound machine (ALOKA) might introduce operator and device-related variability, limiting reproducibility across different settings.]

Response 11: [We fully agree with this comment. Unfortunately, we were not able to perform measurements by two ultrasonographers. However, we will take this suggestion into account in the target study and included it in the study limitations].

Comment 12: [The choice of surgical procedure (VATS or thoracotomy) was determined non-randomly by the surgical team. This lack of randomization could introduce confounding factors related to clinical decision-making, such as tumor size or patient condition.]

Response 12: [Thank you for this comment. Randomized controlled trials (RCTs) are essential for guiding clinical decision-making but are particularly challenging to conduct in surgical fields due to factors such as ethical constraints. In lung cancer surgery, an RCT might result in the abandonment of a beneficial intervention or the adoption of an ineffective one that could potentially harm patients. Therefore, in our study, the surgical method was selected by an experienced surgical team based on the clinical picture and factors such as tumor size, patient age, pulmonary function, and the patient’s overall condition.]

Comment 13: [Patients undergoing bilobectomy, segmentectomy, or pneumonectomy were excluded, limiting the study's applicability to a broader range of thoracic surgical procedures.]

Response 13: [We agree with your comment. However, we would like to point out that we identified this as a limitation of our study in the discussion section, specifically in the paragraph addressing study limitations.]

Comment 14: [Postoperative data collection was confined to the immediate hospital stay, which misses potential long-term complications or recovery patterns associated with diaphragm dysfunction.]

Response 14: [Thank you for your comment. We agree that patients should undergo longer follow-up after their hospital stay and future studies should take this into consideration. However, according to the literature, postoperative pulmonary complications (PPCs) significantly impact surgical morbidity and mortality, particularly within the first postoperative week. For this reason, in this pilot study we primarily focused on assessing the patient’s condition during their hospital stay.]

Comment 15: [The final sample size of 45 patients is relatively small, reducing statistical power and increasing the risk of type II errors. This limitation is particularly significant given the study's focus on specific subgroups with different types of diaphragm dysfunction.]

Response 15: [Of course, we also agree with this suggestion. In the discussion section, within the paragraph on study limitations, we mentioned that the small sample size reduces the statistical power of our study.]

Comment 16: [Blood gas analysis was conducted only on the morning after surgery. This snapshot approach may not fully capture respiratory changes or complications that develop later during the postoperative period.]

Response 16: [We completely agree with your comment. However, we would like to point out that we identified this as a limitation of our study in the discussion section, specifically in the paragraph addressing study limitations. Future studies should perform daily blood gas analyses to capture potential changes as accurately as possible.

Round 2

Reviewer 2 Report

Comments and Suggestions for Authors

The lower part of the new Figure 2 should be removed. Or presented in another way. It is not an incidence if we do not have an absolute number of patients on the graph. Also, from the part of the figure above, the number of patients with these complications is already known.

After Table 2 it should be at least one blank line before the text of the manuscript.

In Tables 2 and 3 PCC as an abbreviation should be added in legends below the tables.

Disfunctions should be written in 1 decimal point. And that one decimal point is of questionable accuracy because the number of patients in groups is less than 100.

In these tables smaller font should be used because the last number of the values is formatted as a single digit belowž

In the discussion section and conclusion, it should be discussed whether there are potential preoperative interventions that could reverse muscle atrophy or increase diaphragmatic muscle thickness.

The statement in the conclusion of the abstract, ''These findings provide valuable insights for predicting and screening high-risk patients for PPCs 30 after lobectomy for lung cancer,'' should be removed. It does not add to new insights. Also, the authors found that expiratory but not inspiratory thickness is associated with postoperative complications. That is the main conclusion of the study, which should be included in the conclusion section.

Author Response

Dear Reviewer,

First, we would like to sincerely thank the reviewer for re-evaluating our manuscript and providing further suggestions to enhance our work and its overall quality. Below, we present our responses to the questions raised.

[Comment 1]: The lower part of the new Figure 2 should be removed. Or presented in another way. It is not an incidence if we do not have an absolute number of patients on the graph. Also, from the part of the figure above, the number of patients with these complications is already known.

[Response 1]: Thank you for this comment. The part of the figure you are referring to corresponds to the old Figure 2, which is crossed out and located below the new Figure 2. It is not included in the corrected manuscript but remains visible due to the requirement to track changes in the text.

[Comment 2]:
After Table 2 it should be at least one blank line before the text of the manuscript.

[Response 2]: We have added a space between the table and the next line of text.

[Comment 3]: In Tables 2 and 3 PCC as an abbreviation should be added in legends below the tables.

[Response 3]: We have included an explanation of the PPCs abbreviation in the tables 2 and 3 legend.

[Comment 4]: Disfunctions should be written in 1 decimal point. And that one decimal point is of questionable accuracy because the number of patients in groups is less than 100.

[Response 4]: As suggested, we have provided values to one digit after the decimal point.

[Comment 5]: In these tables smaller font should be used because the last number of the values is formatted as a single digit below

[Response 5]: We improved the font size in the table. However, file formatting may cause the text to not be on one line.

[Comment 6]: In the discussion section and conclusion, it should be discussed whether there are potential preoperative interventions that could reverse muscle atrophy or increase diaphragmatic muscle thickness.

[Response 6]: Thank you for this comment. We would like to point out that we have already indicated the possibilities of preoperative therapeutic interventions in the conclusion chapter. In our opinion, breathing exercises aimed at strengthening the diaphragm may be an important element of prehabilitation for patients before a planned procedure.

[Comment 7]: The statement in the conclusion of the abstract, ''These findings provide valuable insights for predicting and screening high-risk patients for PPCs 30 after lobectomy for lung cancer,'' should be removed. It does not add to new insights. Also, the authors found that expiratory but not inspiratory thickness is associated with postoperative complications. That is the main conclusion of the study, which should be included in the conclusion section.

[Response 7]: As suggested, we have revised the sentence in the abstract summary.

Round 3

Reviewer 2 Report

Comments and Suggestions for Authors

In Table 5 in the ''Abbreviations''

PaCo2 = partial pressure of carbon dioxide, ''PaCO2'' should be correctly written. Please refer to the journal propositions. The same goes for PaCO2 and HCO3. Maybe the journal uses subscript.